# "Some missions can be quite emotionally painful." Paramedic´s experience exercising coercion during assignments—A qualitative study

Anne Kristine Bergem[1], Nina Øye Thorvaldsen[1], Kristin Häikiö[1]*, Heming Olsen-Bergem[2]

**1** Dept of Nursing and Health Promotion, Faculty of Health Sciences, Oslo Metropolitan University, Oslo, Norway, **2** Dept of Oral Surgery and Oral Medicine, Institute for Clinical Dentistry, Oslo University, Oslo, Norway

* haikio@oslomet.no

## Abstract

### Background

More than 70% of respondents in a previous survey among paramedics reported use of coercion or physical force towards patients. Coercion outside hospital is not permitted, and neither routines nor equipment intended for physical restraint is available in the Norwegian ambulance services. Paramedics carry out assignments involving use of force and coercion on unclear legal grounds, with no training in techniques or proper equipment. Attitudes and experiences of healthcare workers regarding incidents involving coercion in mental health care services are frequently reported in the research literature, yet little is known about paramedics' experiences, and which factors contribute to their moral stress.

### Methods

In the period June-August 2021, almost 400 employees in the ambulance services in a county in the eastern part of Norway were invited to answer a digital questionnaire. One question had an open text field with the question "Can you say something about how you experience transporting patients where force has to be used to secure the patient during transport?". The answers were analyzed using Graneheim and Lundman's content analysis.

### Results

We received eighty-five completed responses (response rate 21%). Force was used by 62 paramedics. Twenty-three left the text field open. The answers showed many unique responses. Content analysis resulted in two overarching themes; 1) lack of routines, equipment, and training regarding use of coercion and force in the ambulance service, and 2) paramedics were confronted with ethical dilemmas, alone and without support from legislation or management.

**Data Availability Statement:** Data cannot be shared publicly because of the risk of recognition of participants in qualitative data. Data are available from the OsloMet Privacy Ombudsman (contact via

personvernombud@oslomet.no) for researchers who meet the criteria for access to confidential data.

**Funding:** The authors received no specific funding for this work.

**Competing interests:** The authors have declared that no competing interests exist.

## Conclusions

The paramedics experienced discomfort related to the exercise of force and coercion during ambulance assignments due to the experience of unclear legislation, lack of training, routines, and equipment in addition to frequent ethical dilemmas and the concern about lack of support from the employer. A clearer legal basis, adapted equipment in the ambulance and regular training, will contribute to greater security in the performance of the work, which will provide safer and more caring treatment for the patients and less moral stress for the staff. With established routines, the employer will be implicitly obliged, and paramedics will be safer in the performance of their work. Ethical reflection must be offered and put into a system.

## Background

The Norwegian Health and Care Services Act [1] defines coercion and force as "measures which the service user opposes, or which are so intrusive that regardless of resistance, they must be considered as use of coercion or force". Coercion is relatively common in mental health care internationally and at the same time considered problematic [2]. In our survey among ambulance workers in a Norwegian county, coercion is defined as the use of physical force. Over 70% of respondents in the survey reported to have exercised such physical force [3]. In health services, coercion is used to ensure proper services for patients who do not have the competence to consent, and to prevent harm to patients or staff [4–8]. In clinical practice, the necessity of using clinical judgment will be present, as described in guidelines from the Norwegian Directorate of Health [9, 10].

The Health Personnel Act gives health personnel the possibility and a duty to provide healthcare without the consent of the patient in acute situations, even if the patient objects to the health care. The provision authorizes the use of coercion in that it states that the duty to help also applies if the patient objects to healthcare [9].

In the health services, the coercion that people are subjected to will be initiated and carried out by other people, i.e., health personnel. Exercising coercion is shown to be morally stressful for health personnel [11]. Moral stress is defined as having to make difficult choices that go against one's own values or the experience of ethical dilemmas and is a risk factor for impaired mental health [12–14].

The exercise of coercion threatens health personnel's understanding of proper care and treatment. Exercising coercion at the same time as being required to provide health care is by Hem et al. [15] described as "a moral enterprise ", i.e., a demanding assignment that is experienced as morally stressful [11]. The main cause of the perceived stress is the conflict linked to the need to maintain control over a situation and the importance of a good therapeutic relationship with the patient. Although healthcare personnel experience coercion as part of their job, coercion is also described as destructive [16]. The balance between taking care of vulnerable people with a mental disorder and society's demands for protection from people with a risk of violence linked to a mental disorder causes moral stress [17].

Personnel in mental health care exercise physical coercion against people already admitted for investigation or treatment of a serious mental disorder in accordance with §4 in the Norwegian Mental Health Protection Act. Coercion is exercised by a legal decision, it is documented and recorded, and all coercive decisions can be appealed by the patient and next of kin [5].

As a general principle, physical coercion is a task reserved for law enforcement agencies. However, if coercion or force is necessary to provide health care of vital importance, the coercion is authorized in the obligation to provide emergency medical care according to Section 7 [19] of the Norwegian Health Personnel Act. Ambulance services, being at the front of health care, often encounter situations where use of coercion is necessary to provide healthcare. However, Norwegian regulations have not been specifically designed, developed, or discussed in the context of the prehospital setting [3]. The Norwegian Penal Code section 17 and 18 provides for actions deemed necessary to save lives or protect others from harm [18]. If coercion or force is necessary to provide health care, the coercion is authorized in the obligation to provide emergency medical care according to Section 7 [19] of the Norwegian Health Personnel Act. Paramedics must therefore have in-depth knowledge of legislation, and the legal basis for enforcement may be unclear.

In hospital wards approved for involuntary mental health care, the Norwegian Mental Health Protection Act [5] applies. Chapter 4 discusses the use of mechanical restraints. Although for many years there has been a strong desire to reduce the use of coercion in mental health care [20], mechanical restraints are available in most inpatient wards. Employees in mental health care departments receive training in the use of mechanical restraints and in many cases have regular training on their use, and the same applies to employees in ambulatory emergency teams in mental health care [21, 22].

As exercise of coercion outside hospital is not permitted, routines have not been developed for the implementation of such measures in the ambulance service. There is no available equipment intended for physical restraint in Norwegian ambulances.

Although it is stated in the description of the paramedic course "As a paramedic you are the extended arm of the hospital and work independently with sick and injured patients", the exercise of coercion and force is not on the educational plans [23].

Paramedics today must carry out assignments that involve the use of force and coercion on unclear legal grounds, outside existing routines, with no training in techniques. Neither do they have access to proper equipment. Together with the high incidence of coercion on ambulance missions [3], it would be reasonable to believe that many paramedics in the Norwegian ambulance service experience participation in "moral enterprises " [15].

Although the attitudes and experiences of healthcare workers regarding situations involving coercion in mental health care services are frequently reported in the research literature [24, 25], little is known about paramedics' experience of exercising coercion [26], and which factors contribute to experiencing moral stress.

## Material and methods

### Inclusion and exclusion criteria

Inclusion criteria: In the period June–August 2021, almost 400 employees in the ambulance service in a county in the eastern part of Norway were invited to answer a digital questionnaire. Included were all ambulance personnel. There were no exclusion criteria. Information letters and requests were sent via e-mail. No information about the participants was collected, and all who worked as paramedics were eligible participants.

### Questionnaire

A digital questionnaire, Nettskjema [27] was sent as an open link to all who wanted to participate. The questions asked were about the paramedics' exercise of power and coercion. One question with an open text field read: " Can you say something about how you experience transporting patients where force has to be used to secure the patient during transport?" The

question is not a validated one but was made based on the second author´s experience from the ambulance service and tested out among a group of test persons by an area instructor for paramedics in the county where the study later was performed.

## Ethical considerations

Participation was voluntary, and the survey was anonymous. The Norwegian Center for Research Data (NSD) assessed the data (ref. no. 775741). Regional committees for medical and health-related research ethics (REK) department south-east C assessed the purpose of the study as outside the Health Research Act (submission assessment ref. no. 270172).

## Trustworthiness

The open field question answered in this study, although not validated due to the novelty of the study topic, is open and a part of a larger survey. A test group of paramedics described the question as easily understood. No misinterpretation was detected among the test persons or area instructor.

## Analysis

The answers in the open text fields were analyzed using Graneheim and Lundman's analysis of qualitative content [28].

In accordance with Graneheim and Lundman, we have identified both a manifest and a latent content of the text. The manifest content is based on the visible, obvious, and verbatim written by the informants. In the latent content, we interpret the underlying meaning in the text and raise the meaning in the text to a higher level of abstraction through interpretations and theme formation. The first and last author read through each text response several times first separately and then together, to get an overall picture of the content. Disagreements would have been solved by the other authors. The analysis was conducted in five steps. Step one: In collaboration, the meaningful elements were extracted from the answers in accordance with the purpose of the study. Meaningful elements are groupings of words that relate to the same meaning. These elements were during step two reduced to a condensed text making effort not to change the core of the message [28]. In step three, an abstraction was carried out, that is, an interpretation of the condensed meaningful elements. In step four, the following question was posed to each condensed element: "What is this really about?" This resulted in several codes that were assessed against each other to find similarities and differences. The codes are short descriptions of the content of a group of meaningful elements with approximately the same content. In step five, the codes were sorted into categories based on their topic. The categories consist of thematically similar codes [28]. Table 1 shows an example of the analysis process from meaningful unit to theme.

**Table 1. Example of content analysis.**

| Meaningful elements | Condensed element of meaning | Code | Category | Theme |
|---|---|---|---|---|
| *Experience it as insufficient both in relation to procedural work and available equipment.* | It feels unsafe both in terms of procedures and equipment to exercise coercion. | Unclear rules Practical problem | There are practical problems and missing routines for my work. | Lack of routines, equipment, and training regarding the use of coercion and force in the ambulance service. |

When translating the quotes into English, the context for the statements was attached, as well as the interpretation made through the content analysis.

## Results

A total of 85 completed responses from paramedics were received, which corresponds to a response rate of 21%. Sixty-two persons (72.9%) had used force to ensure that patients were properly secured during transport. 23 of the respondents left the text field open. In the open text answers, we found many unique responses. Content analysis ad modum Graneheim and Lundman, however, resulted in two overarching themes. The two themes were 1) lack of routines, equipment, and training regarding use of coercion and force in the ambulance service, and 2) Paramedics are often faced with ethical dilemmas, alone and without support from legislation or management.

The quotes are from different paramedics. Names and sex are fictive.

### Lack of routines, equipment, and training regarding use of coercion and force in the ambulance service

In many of the responses, paramedics described the practical challenges they experience. Lack of standardized suitable equipment, such as conveyor belts, and lack of proper training, makes it necessary to invent adaptive and individualized solutions from job to job.

> Quote: *I wish we had suitable equipment like the mechanical restraints used in psychiatric wards, I feel it would make the assignments that require restraining the patient easier to manage.* (Barry)

The respondent above, describes the coercion required as difficult to perform due to lack of suitable mechanical equipment, and this was supported by many other participants. As the next quote illustrates, respondents also request a plan for safe and secure transportations of people who need restrains.

> Quote: *Sometimes I think that; when we know that these transports take place, why can't we also make plans for it to be conducted in a safe and secure way with mechanical restraints in place so that we don´t have to physically hold the patient. Can we not practice and be good at using mechanical restraints instead of putting ourselves, the police and the patient at risk by tumbling around in the back of an ambulance.* (Greg)

This respondent highlights the challenge that arises from the fact that use of coercion in the ambulance service only is authorized if the healthcare is of vital importance or to protect the patient or others from harm. However, planning and training for the use of coercion regulated by Section 17 in the Penalty Code is not permitted, even if the incidence is high. The respondent compares his/her tasks with the tasks of health personnel in mental health care and points out that the same work tasks are carried out under very different conditions, legally and practically.

In addition to necessary equipment and a plan for emergency situations, several participants requested more knowledge and training related to use of coercion. The next quote illustrates this.

> Quote: *I think we have too little knowledge to use coercion and force and find it unpleasant to have to sort things out ourselves because you don't get the necessary assistance.* (Paul)

In the quote above and below, several aspects of coercion are mentioned. Knowledge in this context is interpreted as a lack of training, and the respondent, like many other respondents, mention that the use of coercion against patients is experienced as unpleasant and with little

support from other resources It may also appear that the respondent not necessarily thinks increased knowledge is needed, and instead emphasizes lack of knowledge to be compensated with necessary assistance.

> Quote: *We need more practical training, around different situations that can arise in connection with this—i.e., how best to handle it.* (Iris)

Several respondents call for specific training in exercising coercion.

## Paramedics are often faced with ethical dilemmas, alone and without support from legislation or management

In many replies, despair is expressed as a response to exposing people to coercion, even though it is for their own benefit. The respondent expresses empathy with the patient, as well as concern that he/she, as a paramedic, imposes new burdens on patients in situations where they need help. This is illustrated by one of the respondents in the next quote:

> Quote: *Often these people are patients who have had their problems due to abuse in the past. The fact that we use coercion feels like another trauma.* (Linda)

Several respondents express the use of coercion as necessary for the ambulance mission to be carried out. In several responses, it appears paramedics are feeling safe when the police are present, both because the police are the ones who exercise the coercion, excusing the paramedics, and because the paramedics do not have to decide whether they are allowed to exercise coercion.

> Quote: *I think the police are the ones who should exercise force and all such transports should take place in a police car.* (Sam)

Several responders express concerns about having the sole responsibility and it appears that the exercise of force and coercion is not perceived as part of what ambulance workers are educated and trained for. Several respondents do not experience the exercise of force and coercion as a task for healthcare personnel but designates the police as the right authority, like in the quote below.

> Quote: *I think it is as a mistake. It is not our job to force people.* (Rob)

A few of the respondents were expressing concern about lack of support from management if the use of coercion were to be complained about or otherwise known to the public, like in the quote below.

> Quote: *Uncomfortable with doubts about the management having my back covered.* (Terry)

The answer is an example of the fear of making a mistake, i.e., exercising coercion and power without sufficient legal basis, and that the actions performed will be regarded as something the person has done without the management´s approval.

## Discussion

The results show that, for several reasons, paramedics experience the exercise of coercion and force during assignments as unpleasant, even though many express the use of force as

necessary. The results are in accordance with experiences from personnel who work with municipal health work and in mental health care [11, 15–17, 29]. Both practical aspects of coercion linked to a lack of training and equipment, and the more emotional and legal aspects are problematized by the respondents.

Although the exact extent of the exercise of force and coercion during ambulance assignments is not known, and our material is limited to one region, there are many indications that the incidence is so high that it does not make sense to talk about rarity, coincidence, or one-off events. A national survey with the same questionnaire shows the same trend [30].

## Lack of support in legislation and management

"Not our job" and "should take place in a police car" are statements that exemplify paramedics´ doubts about their own role in the exercise of force and coercion. The lack of a clear legal basis in health legislation is perceived as problematic.

Paramedics must use force without being sure whether they are allowed to do so. For that reason, paramedics ask for assistance from the police when the patient resists healthcare, even when the paramedics are physically able to handle the patient's resistance. The police are thus asked for assistance for missions where coercion is necessary, even if in practice police assistance is unnecessary [31].

In Poland, EMS personnel were allowed to exercise coercion in 2010. Each incident is documented in a separate record system, and it is argued that both the patients' rights as well as those of healthcare personnel have been strengthened following the change in the law [32].

Without routines regarding documentation, there is a risk that exercised coercion is not documented.

## Routines, equipment, and training

Since training and education in healthcare must be in line with then legislation regulating the service, one cannot train and plan for emergency use of coercion in the prehospital health care services. In mental healthcare, routines for exercising coercion are established [20, 21]. For the same reason, paramedics do not have suitable equipment developed for securing patients during transport which would be preferable using bandage materials etc. which are used today [3].

Education and training must include prevention and ethical reflections on one's own exercise of force and coercion. For employees in mental health care, this is already well described [33]. In the training manual developed for mental health care, MAP (Encounter with aggression problems), it is expressed as follows: "Furthermore, the concept of care includes our view of people and our attitudes when dealing with the individual" [34]. When use of coercion in the ambulance services is not recognized, lack of practical training as well as lack of ethical reflections will be a consequence.

## Moral enterprise–ethical dilemmas

It is beyond the scope of this article to discuss the experience of people who are subjected to coercion and force, but as the responses in the survey show, paramedics reflect on the force and coercion they subject other people to and recognize that it can be harmful. The recognition is necessary and important to provide good care at the same time as they reinforce the ethical dilemmas for each individual paramedic.

Without support in legislation or management, handling the burden by exercising coercion is left to each individual employee, or each pair of paramedics.

When the use of coercion and force occurs frequently, and paramedics experience discomfort during the execution, this means that every week there are many people who experience

moral stress in their normal work, and thus have an increased risk of mental health problems [13, 14].

A clearer legal basis will remove some uncertainty about legality. Better cooperation with the police will create a better distribution of work and tasks, but the ethical dilemmas and the emotional discomfort associated with exercising force and coercion will not and should not be removed by expanding the legislation.

It is therefore crucial that employers and organizations take greater responsibility for having better support systems and structures to reduce moral stress. Kälvemark et al. suggests more education in ethics and forums to discuss ethically difficult situations or hospital personnel [14].

Based on the findings in our study, there is a need of closer cooperation between police and ambulance personnel both for practice and training but also regarding the development of new guidelines. Ethical discussions regarding the use of force must be introduced as early in education and training as possible, and psychosocial support on the work place should be encouraged.

## Strengths and weaknesses of the study

It is a strength that the survey includes many respondents, albeit from a limited geographical area. The pilot study has been followed up by a national study, which shows the same tendency [30]. A disadvantage is that none of the answers go in-depth. Interviews would have given the opportunity for more in-depth answers. A strength of an open question in a form is that the preconceptions of the authors have not influenced the answers given. The fact that the first author is a psychiatrist herself and has known the burden of making decisions about the use of coercion in mental health care may have influenced the interpretation of the results in the analysis process.

## Conclusion

Paramedics in the county we researched, experience discomfort related to the exercise of force and coercion during ambulance assignments. The discomfort is due to the experience of unclear legislation, lack of training, routines, and equipment in addition to frequent ethical dilemmas and the concern about lack of support from the employer. A clearer legal basis, more adapted equipment in the ambulance and regular training in situations where it is necessary to use force, will contribute to greater security in the performance of the work, which both provides safer and more caring treatment for the patients and less moral stress for the staff. With established routines, the employer will be implicitly obliged, and paramedics will be safer in the performance of their work. In addition, ethical reflection must be offered and put into a system.

## Acknowledgments

The authors acknowledge Nils Halvor Gryting for contributing to the distribution of the survey.

## Author Contributions

**Conceptualization:** Anne Kristine Bergem, Nina Øye Thorvaldsen, Kristin Häikiö.

**Data curation:** Kristin Häikiö.

**Formal analysis:** Anne Kristine Bergem, Heming Olsen-Bergem.

**Investigation:** Anne Kristine Bergem, Nina Øye Thorvaldsen.

**Methodology:** Anne Kristine Bergem, Nina Øye Thorvaldsen, Kristin Häikiö.

**Project administration:** Anne Kristine Bergem, Nina Øye Thorvaldsen, Kristin Häikiö.

**Supervision:** Kristin Häikiö, Heming Olsen-Bergem.

**Writing – original draft:** Anne Kristine Bergem, Heming Olsen-Bergem.

**Writing – review & editing:** Nina Øye Thorvaldsen, Kristin Häikiö, Heming Olsen-Bergem.

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
