## [Decision Letter · Decision Letter 0]

4 Oct 2023

PONE-D-23-25000"Some missions can be quite emotionally painful."

Paramedic´s experience of exercising coercion during assignments-a qualitative study.PLOS ONE

Dear Dr. Häikiö,

Thank you for submitting your manuscript to PLOS ONE. After careful consideration, we feel that it has merit but does not fully meet PLOS ONE’s publication criteria as it currently stands. Therefore, we invite you to submit a revised version of the manuscript that addresses the points raised during the review process.

We look forward to receiving your revised manuscript.

Kind regards,

Collins Atta Poku

Academic Editor

PLOS ONE

Reviewers' comments:

Reviewer's Responses to Questions

**Comments to the Author**

1. Is the manuscript technically sound, and do the data support the conclusions?

Reviewer #1: Yes

Reviewer #2: Yes

2. Has the statistical analysis been performed appropriately and rigorously? 

Reviewer #1: Yes

Reviewer #2: N/A

3. Have the authors made all data underlying the findings in their manuscript fully available?

Reviewer #1: Yes

Reviewer #2: Yes

4. Is the manuscript presented in an intelligible fashion and written in standard English?

Reviewer #1: Yes

Reviewer #2: Yes

5. Review Comments to the Author

Reviewer #1: Paper is an interesting read and presents valuable information which will benefit paramedics and other health professionals.

A few suggestions: on page 11 "... Norwegian Mental Health Protection Act (4) applies. Chapter 4..." the point on chapter 4 should be clearer so that readers know the Act is being referenced. It could be written as "Chapter 4 of the Act..."

The Discussion from page 17 can be made richer.

On the whole manuscript is a good read

Reviewer #2: The authors present a study examining the experiences of paramedics regarding the use of coercion in pre-hospital care. The study is interesting and contextually situated with implications for not only health services, but also legislation/ law and ethics. The authors are to be commended for undertaking a study in such a sensitive area that may evoke intense emotions. That said, there are areas of the study that warrants further attention. Please see my comments below for your kind consideration:

Background:

1. Paragraphs in the background can be developed further. Presenting three lines as a standalone paragraph can be improved further. For instance the second and third paragraphs can be merged and expanded further on.

2. The authors do an excellent work at presenting the state-of-the-art Norwegian literature in the background, but I believe it will be helpful if the authors move a step further to consider the global context as well to ground their study for international readership.

3. The literature on moral stress can be expanded further. For instance, does the patient have the right to refuse treatment? At what point does the patient's right end for coercion to be used? Are there any regulations that protect the responsibility of the paramedics to use coercion? A deeper engagement in this area with more relevant ethical principles will be of great help to readers.

Materials and methods:

1. It will be of great help if the authors use applicable sub-headings here. Putting all the details under one topic may not be helpful to readers. For instance, inclusion and exclusion criteria are not mentioned. The authors do mention that all paramedics were eligible, but these details need to be made more explicit.

2. The data analysis component is really well presented, but can be made to stand alone.

3. There is no mention of trustworthiness or rigor in this study; this is an essential component of studies employing qualitative designs. Please add a section on this.

4. From the ethical perspective, it will be helpful for the authors to highlight how security and confidentiality were ensured using the digital questionnaire? Also, how were the questions drafted? What sampling approach was used? How did the authors ensure diversity?

Results

1. The use of content analysis is appropriate for this study, but the mention of themes creates confusion as that is more aligned towards thematic analysis. Content analysis usually proceeds from codes, sub-categories, and categories (rather than themes). Besides, it is confusing to talk about overarching themes in a study that employed content analysis. Please rectify this.

2. If possible, a table showing the sub-categories and categories should be presented. Additionally, with the use of the content analytical approach, it will be helpful if the authors indicate how many participants exemplars are congruent with a category.

3. The participants' demographic information are not included? At least a summary is needed to giver readers an overview of who the participants are.

4. Regarding the presentation of the study findings, it is difficult to know which ones are the main categories and the sub-categories. I will suggest the authors present the main categories with a high level summary before the sub-categories if possible.

Discussion

1. The discussion section is well raised, well done to the authors. It may be helpful to draw the policy implications of the study considering the complex interplay of legislation, legal, ethical, and health issues.

Many thanks to the authors once again, for this interesting study. I look forward to the revised manuscript and published version of the study soon.

6. PLOS authors have the option to publish the peer review history of their article (what does this mean?). If published, this will include your full peer review and any attached files.

Reviewer #1: **Yes: **Priscilla Yeye Adumoah Attafuah

Reviewer #2: **Yes: **Jonathan Bayuo

---

## [Author Response · Author response to Decision Letter 0]

14 Nov 2023

Reviewer #1: The paper is an interesting read and presents valuable information that will benefit paramedics and other health professionals.

A few suggestions: on page 11 "... Norwegian Mental Health Protection Act (4) applies. Chapter 4..." the point on Chapter 4 should be clearer so that readers know the Act is being referenced. It could be written as "Chapter 4 of the Act..."

Thank you for pointing out the need for specifications. We have added the specification.

The discussion from page 17 can be made richer.

Thank you for encouraging us to elaborate on the findings of our study. We have added a paragraph to the discussion.

On the whole manuscript is a good read.

Our sincere thanks for this very nice comment.

Reviewer #2: The authors present a study examining the experiences of paramedics regarding the use of coercion in pre-hospital care. The study is interesting and contextually situated, with implications for not only health services but also legislation, law, and ethics. The authors are to be commended for undertaking a study in such a sensitive area that may evoke intense emotions. That said, there are areas of the study that warrant further attention. Please see my comments below for your kind consideration:

Background:

1. Paragraphs in the background can be developed further. Presenting three lines as a standalone paragraph can be improved further. For instance, the second and third paragraphs can be merged and expanded further.

Thank you for your comment. We have merged the two paragraphs and expanded according to the reference.

2. The authors do excellent work at presenting the state-of-the-art Norwegian literature in the background, but I believe it will be helpful if the authors move a step further to consider the global context as well to ground their study for international readership.

Thank you for your comment. We have added some international perspectives in the background and added one reference (2).

3. The literature on moral stress can be expanded further. For instance, does the patient have the right to refuse treatment? At what point does the patient's right end for coercion to be used? Are there any regulations that protect the responsibility of paramedics to use coercion? A deeper engagement in this area with more relevant ethical principles will be of great help to readers.

Thank you for your comment. A section is added to elaborate on the use of coercion and patient´s rights.

Materials and methods:

1. It will be of great help if the authors use applicable sub-headings here. Putting all the details under one topic may not be helpful to readers. For instance, inclusion and exclusion criteria are not mentioned. The authors do mention that all paramedics were eligible, but these details need to be made more explicit.

Thank you for your comment. We have made three subheadings and clarified according to your suggestions.

2. The data analysis component is really well presented but can be made to stand alone.

Thank you for your suggestion. We have changed the analysis part accordingly.

3. There is no mention of trustworthiness or rigor in this study; this is an essential component of studies employing qualitative designs. Please add a section on this.

Thank you, a section has been added.

4. From an ethical perspective, it will be helpful for the authors to highlight how security and confidentiality were ensured using the digital questionnaire? Also, how were the questions drafted? What sampling approach was used? How did the authors ensure diversity?

Thank you. We may not have been clear enough on this, but there was only this one open question, and it was at the end of a quantitative survey. Regarding security and confidentiality, no information about the participants was collected. All data was anonymous throughout the process. We believe this to be clearly stated in the manuscript under Material and Methods, in the new paragraph “Inclusion and Exclusion Criteria," p.6.

Results

1. The use of content analysis is appropriate for this study, but the mention of themes creates confusion as that is more aligned towards thematic analysis. Content analysis usually proceeds from codes, sub-categories, and categories (rather than themes). Besides, it is confusing to talk about overarching themes in a study that employed content analysis. Please rectify this.

Thank you for your comment. As themes are a part of this method of analysis, described by Graneheim and Lundman, we choose to uphold our nomenclatur as it is. 

2. If possible, a table showing the sub-categories and categories should be presented. Additionally, with the use of the content analytical approach, it will be helpful if the authors indicate how many participants exemplars are congruent with a category.

Thank you for your comment. Again, subcategories are not used in the Graneheim and Lundman analysis. This content analysis uses the process: meaningful elements – condensed elements of meaning – codes – categories ,– themes. We have chosen to follow Graneheim and Lundman's analysis of qualitative content. Please find the description on p 7 under “Analysis”. Changing this will require doing the analysis process all over again.

3. The participants' demographic information are not included? At least a summary is needed to giver readers an overview of who the participants are.

Thank you for your comment. We were asked by the editor to remove which area in Norway the participants come from. Other demographics were not collected due to GDPR.

4. Regarding the presentation of the study findings, it is difficult to know which ones are the main categories and the sub-categories. I will suggest the authors present the main categories with a high level summary before the sub-categories if possible.

Thank you for your comment. See answer given under question 2 (Results).

Discussion

1. The discussion section is well raised, well done to the authors. It may be helpful to draw the policy implications of the study considering the complex interplay of legislation, legal, ethical, and health issues.

Thank you for encouraging us to be more specific about the implications of our study. We have elaborated on the discussion with an additional paragraph.

Many thanks to the authors once again, for this interesting study. I look forward to the revised manuscript and published version of the study soon.

---

## [Decision Letter · Decision Letter 1]

10 Dec 2023

"Some missions can be quite emotionally painful."

Paramedic´s experience exercising coercion during assignments—a qualitative study

PONE-D-23-25000R1

Dear Dr. Häikiö,

We’re pleased to inform you that your manuscript has been judged scientifically suitable for publication and will be formally accepted for publication once it meets all outstanding technical requirements.

Kind regards,

Collins Atta Poku

Academic Editor

PLOS ONE

Additional Editor Comments (optional):

Reviewers' comments:

Reviewer's Responses to Questions

**Comments to the Author**

1. If the authors have adequately addressed your comments raised in a previous round of review and you feel that this manuscript is now acceptable for publication, you may indicate that here to bypass the “Comments to the Author” section, enter your conflict of interest statement in the “Confidential to Editor” section, and submit your "Accept" recommendation.

Reviewer #2: All comments have been addressed

2. Is the manuscript technically sound, and do the data support the conclusions?

Reviewer #2: Yes

3. Has the statistical analysis been performed appropriately and rigorously? 

Reviewer #2: N/A

4. Have the authors made all data underlying the findings in their manuscript fully available?

Reviewer #2: Yes

5. Is the manuscript presented in an intelligible fashion and written in standard English?

Reviewer #2: Yes

6. Review Comments to the Author

Reviewer #2: Many thanks to the authors for thoughtfully addressing all the comments raised. No further comments from me.

7. PLOS authors have the option to publish the peer review history of their article (what does this mean?). If published, this will include your full peer review and any attached files.

Reviewer #2: No

---

## [Editor Report · Acceptance letter]

20 Dec 2023

PONE-D-23-25000R1 

PLOS ONE

Dear Dr. Häikiö, 

I'm pleased to inform you that your manuscript has been deemed suitable for publication in PLOS ONE. Congratulations! Your manuscript is now being handed over to our production team.

Kind regards, 

on behalf of

Dr. Collins Atta Poku 

Academic Editor

PLOS ONE